# Wearable Health Technology for Preoperative Risk Assessment in Elderly Patients: The WELCOME Study

**DOI:** 10.3390/diagnostics13040630

**Published:** 2023-02-08

**Authors:** Massimiliano Greco, Alessandra Angelucci, Gaia Avidano, Giovanni Marelli, Stefano Canali, Romina Aceto, Marta Lubian, Paolo Oliva, Federico Piccioni, Andrea Aliverti, Maurizio Cecconi

**Affiliations:** 1Department of Biomedical Sciences, Humanitas University, 20072 Milan, Italy; 2Department of Anesthesia and Intensive Care, IRCCS Humanitas Research Hospital, 20089 Milan, Italy; 3Dipartimento di Elettronica, Informazione e Bioingegneria, Politecnico di Milano, 20133 Milan, Italy; 4META—Social Sciences and Humanities for Science and Technology, Politecnico di Milano, 20133 Milan, Italy; 5Clinical Engineering, IRCCS Humanitas Research Hospital, 20089 Milan, Italy

**Keywords:** wearable electronic devices, fitness trackers, perioperative care, risk assessment, exercise test, general surgery, anesthesia

## Abstract

Preoperative identification of high-risk groups has been extensively studied to improve patients’ outcomes. Wearable devices, which can track heart rate and physical activity data, are starting to be evaluated for patients’ management. We hypothesized that commercial wearable devices (WD) may provide data associated with preoperative evaluation scales and tests, to identify patients with poor functional capacity at increased risk for complications. We conducted a prospective observational study including seventy-year-old patients undergoing two-hour surgeries under general anesthesia. Patients were asked to wear a WD for 7 days before surgery. WD data were compared to preoperatory clinical evaluation scales and with a 6-min walking test (6MWT). We enrolled 31 patients, with a mean age of 76.1 (SD ± 4.9) years. There were 11 (35%) ASA 3–4 patients. 6MWT results averaged 328.9 (SD ± 99.5) m. Daily steps and 𝑉𝑂2𝑚𝑎𝑥 as recorded using WD and were associated with 6MWT performance (R = 0.56, *p* = 0.001 and r = 0.58, *p* = 0.006, respectively) and clinical evaluation scales. This is the first study to evaluate WD as preoperative evaluation tools; we found a strong association between 6MWT, preoperative scales, and WD data. Low-cost wearable devices are a promising tool for the evaluation of cardiopulmonary fitness. Further research is needed to validate WD in this setting.

## 1. Introduction

As a result of the global increase in life expectancy and chronic comorbidities, the number of complex procedures involving elderly and frail patients will rise [1,2]. Despite advancements in surgical and anesthetic techniques, cardiovascular and perioperative complications continue to be a burden, even in settings with abundant resources [3]. Considerable research effort has been devoted to the preoperative identification and stratification of high-risk populations to improve perioperative outcomes.

To allow an estimation of risks, several scales are generally employed, including the American Society of Anesthesiology Physical Status (ASA Physical Status) [4,5], Clinical Frailty Scale (CFS) [6], Metabolic Equivalent of Task [7], and Duke Activity Status Index (DASI) [8], which allow for a quick and standardized patient assessment, with the limit of subjectivity and poor predictivity. Specific tests exit to investigate the functional status, including the gold standard cardiopulmonary exercise testing (CPET), which is expensive and time consuming and therefore reserved for selected cases [9]. The Six Minute Walking Test (6MWT) has been recognized as an inexpensive and quick fitness assessment method associated with a good prediction of mortality, especially for chronic obstructive pulmonary disease, chronic heart failure, and pulmonary arterial hypertension [10,11,12,13].

Wearable devices are new tools that are gaining attention in the medical field, for clinical management both inside and outside hospital settings [14,15,16]. Wearable devices have been shown to be useful for the collecting and monitoring of vital and physiological parameters such as hearth rate (HR), sleep quality, expended calories, and physical activity, and can also be incentivized thanks to programs and features, therefore creating a positive impact on subjects’ everyday life [17,18].

Two main categories exist: medical wearable devices and commercial smart watches, which are now diffuse everywhere and that function mostly as an extension of a subject’s mobile phone, and host several sensors, as accelerometers and photoplethysmography (PPG), to detect physical parameters such as the number of steps, activity, and heart rate. While less precise than medical devices, commercial devices have been used in medical fields as a self-assessment tool [19,20].

A review of the literature revealed a scarcity of studies on the use of wearable commercial devices in the perioperative phase, and to our knowledge, no study has investigated the possibility of employing wearable devices as a complementary tool to improve preoperative evaluation. In the present study, commercial wearable devices (WD) were incorporated into preoperative evaluation in order to test the hypothesis that their data may be associated with results from preoperative scales and exercise testing and may thus be adopted in preoperative risk stratification.

## 2. Materials and Methods

We conducted a prospective observational study, including high-risk patients undergoing surgery, between 1 May 2020 and 31 May 2021. The study was approved by Humanitas Research Hospital Ethical Committee (IRB number 350/21, 20 April 2021) and was registered in clincaltrials.gov (NCT05083598). Written informed consent was collected for all patients. The inclusion criteria were the following: the patients were aged 70 years or older, undergoing a surgical procedure longer than two hours and that was planned to be performed under general anesthesia. Exclusion criteria were inability to express consent, need for urgent or emergency procedures, and patients with limited physical activity/limited mobility related with neurological or orthopedic disease, or recent acute cardiovascular event. Consenting patients were recruited at a preoperatory evaluation clinic and were provided with a commercial wearable device (described below) that they were instructed to wear for 7 consecutive days starting from enrolment, while conducting their normal life during the time window. The primary outcome was to assess whether wearable devices can identify patients at risk of postoperative complications. Secondary outcomes are the association between the device-acquired objective parameters and the evaluation measures of the following clinical scales: 6MWT, Clinical Frailty Scale (CFS), Metabolic Equivalent (MET), and Duke Activity Status Index (DASI). 

Immediately after inclusion and while wearing the device, patients underwent 6MWT as part of a preoperative evaluation. To determine whether patients presented a reduced exercise tolerance, a discriminative threshold of 350 m was chosen for 6MWT, according to the published literature [9]. Respiratory rate, Borg Scale, blood pressure, and heart rate were clinically acquired before and after the completion of the 6MWT in all patients except the first three. On patient enrolment, we conducted a standard anesthesiologic assessment, which included comorbidities, an airways assessment, ASA class, assessment of Metabolic equivalent (MET), Duke Activity Status Index, and Clinical Frailty Scale.

For this study, we chose as the commercial WD the Fitbit Inspire 2, due to its long battery life, which lasts about 7 days without the need for battery recharging, with small dimensions (37 × 17 × 13 mm) and weight (31 g), and its simplicity of use and low costs. Through the Fitbit App, available for Android and iOS, the operator entered the patients’ information on age, sex, height, and weight, without personal identifiers. The memory of the device allows it to store 7 days of detailed data (minute by minute), and daily totals for the previous 30 days. By synchronizing the device with the mobile app, data are uploaded to the cloud and can be downloaded through APIs on the web panel. The patients enrolled in the study did not see the data while they were collected, as data were stored locally on the device and then downloaded by the clinicians at the end of the acquisition. The device is equipped with a 3-axis accelerometer for movement detection, an optical HR monitor (PPG sensor) composed of green and infrared LEDs and PDs and a Bluetooth 4.2 radio transceiver. The HR and activity level data were collected, and from these data, the following parameters were estimated: baseline HR, maximum HR, HR variability, daily and maximum physical activity, caloric expenditure, and HR recovery after stress. In this study, we evaluated the average walked steps per day, maximal oxygen consumption (VO2max, as processed by the Fitbit device), HR data, activity intensity, energy expenditure, and calories. 

In the Inspire 2 device, heart rate is recorded through the device’s PPG, and data are available at different sampling rates, with a distance in the order of seconds one from the other, with a higher sampling rate when exercise is detected. The resting HR is automatically computed on a 24-h basis. We defined the recovery HR as the decrease in HR one minute after exercise cessation.

The number of walked steps and distance covered are recorded through the 3-axis accelerometer, while processing algorithms estimate the cadence. Distance is computed as the number of steps times the stride length (based on sex and height information entered by the user) or can be more accurately estimated if the device is linked to a smartphone (not used in this study). Calories burnt are calculated using a combination of physical activity and HR metrics. Also, a basal metabolic rate is estimated based on information on the characteristics of the subject. Minutes of physical activity were divided into 4 classes based on activity intensity. 

### Statistical Analysis

As this was a pilot study, no formal sample size calculation was performed. Data were extracted at the end of the 7-day study period, synchronizing the device with a computer. More complex features were engineered starting from the extracted data; this process is described elsewhere in a methodological paper (Angelucci A et al., currently under peer-review) [21]. Clinical data and data extracted from the device are here presented as mean and standard deviation (SD), or as frequency/percentage, as appropriate. Correlation was assessed with non-transformed data and after nonlinear transformation (i.e., in case of daily number of walked steps walked). All analyses were conducted in Python. 

## 3. Results

A total of 31 patients were enrolled, 9 (29%) women and 22 (71%) men, with a mean age of 76.1 (SD ± 4.9) years. The flow chart of patient enrolment is reported in Figure 1. 

Baseline data, and results from preoperative clinical evaluation scales are reported in Table 1.

All patients underwent the 6MWT. On average, patients walked 328.9 (SD ± 99.5) m, while 14 (45%) patients could not walk a distance greater than 350 m. Four patients interrupted the test before 6 min due to the onset of moderate symptoms or inability to continue: P06 stopped at the fourth minute, P23 at the third, while P12 and P20 managed to endure the test for just two minutes. The result of the 6MWT for each patient is displayed in Figure 2.

Figure 3 reports the results of MET, DASI, and CFS clinical evaluation scales according to the 6MWT results (above or below 350 m threshold) and to high and low baseline activity as detected by the WD as average daily steps (above or below 10,000 average steps per day). 

As evident from Figure 3, most patients that exceeded the 350 m threshold of the 6MWT also walked more than 10,000 steps per day on average, while only four of this group did not. When logarithmic transformation of the number of steps walked daily is applied, there was good correlation with the 6MWT performance (R = 0.56, *p* = 0.001). Correlation between the logarithmic transformation of daily step count and the 6MWT is reported in Figure 4. As depicted in this figure, the logarithm of the daily step count demonstrated good correlation with the 6MWT results (total walked distance), with r = 0.57 and *p* = 0.001.

The total amount of calories burnt in each day by the patients followed the same trend as daily steps (1594.01 ± 659.05 in patients walking more than 350 m, vs. 1093.95 ± 612.46 calories in patients walking less than 350 m, *p* < 0.001). We report total daily walked steps, 6MWT distance, results from preoperative evaluation scales, and patient characteristics in Table 2 according to 6MWT high or low performance and to WD daily steps above or below the 10,000 steps threshold. 

The average number of minutes that subjects spent doing high-intensity activity as reported by the WD was correlated with the 6MWT results, which was significantly higher in the group walking more than 350 m, even if with large variations among the subjects (26.25 ± 28.21 min vs. 8.93 ± 17.53 min, *p* < 0.001).

Results of level of physical activity as tracked by the WD per each subject are reported in Figure 5. Light physical activity as recognized by the WD was most prevalent for all patients. On the contrary, several but not all patients displayed some levels of moderate and vigorous physical activity.

Considering the patients with positive values, the weekly average of 𝑉𝑂2𝑚𝑎𝑥 as reported by the WD correlated strongly with the results of the 6MWT. As depicted in Figure 6, the maximal oxygen consumption (VO2 max) showed a high correlation with results from the 6MWT (walked distance in meters), with an r = 0.58 and *p* = 0.006.

The HR measured by the device’s sensor showed a significant variability among patients, and no correlation. Several patients reached a HR greater than 160 bpm, while others rarely surpassed 120 bpm. The recovery HR has been normalized and correlated to the distance walked during the 6MWT and showed an intermediate correlation (r = 0.45).

Two patients out of thirty-one did not undergo surgery due to a change in the surgical plan after enrollment and anesthesiologic evaluation. There was a low number of postoperative complications (3 patients, 10.3%) and no correlation was found between postoperative complications and the 6MWT performance or with the WD data. Only 1 patient out of 29 died (3.4%) due to late surgical complications, after a discharge and readmission to hospital.

## 4. Discussion

The WELCOME study is the first study to assess the correlation between data acquired through commercial wearable devices and benchmark clinical tests for preoperative evaluation. In this pilot study, we demonstrated that a simple, low-cost device can be used to track patient home activity over several days before surgery, and that daily steps and VO2 data recorded at home through wearable devices are correlated with the results from the 6MWT performed in hospital settings and with clinical evaluation scales. This study provides a foundation for assessing patients’ functional capacity in the context of preoperative evaluation using low-cost, consumer-grade wearable technology.

We included a population of elderly and frail patients, over seventy years old, undergoing general anesthesia lasting at least two hours. The population was defined to include intermediate and high-risk patients. Hence, the walked distances of these subjects are poor, with 45% of patients not able to walk over the defined threshold of 350 m at 6MWT. 

The correlation between the 6MWT results and average walked steps per day proved to be strong (R = 0.56, *p* = 0.001), and most patients that exceeded the 350 m threshold of the 6MWT also walked more than 10,000 steps per day on average, while only four of this group did not reach the 10,000 steps thresholds. Using data from commercial WD could provide a low-cost alternative to performing in-hospital exercise testing such as the 6MWT, a useful option for patients located at a distance from the hospital, or when hospitals have reduced access policies, as in the case of the recent COVID-19 pandemic. In these settings, wearable devices could be theoretically distributed by a general practitioner and tracked data remotely analyzed at surgical referral centers, saving resources, patients and caregivers’ time. The great difference in number of daily steps in this population may be attributed to the variability among patients, considering age, frailty, and other characteristics impacting on subjects’ everyday life. Some patients had mild motility impairment or mild cognitive impairment, so they were not active in general, but many patients complained of a recent decrease in energy and motivation, often associated to the onset of the disease that was being treated (in most cases lung, liver, or abdominal cancer). However, a number of subjects reported a very active lifestyle, with regular physical activity, and a couple of patients were still actively working, so they were generally more engaged and active during the day compared to others.

Regarding clinical scales, CFS, DASI, and ASA scales were mostly concordant between them, probably due to the inclusion criteria. The CFS score was generally high in these patients, with a mean of 4 in ASA 3–4 patients. Similarly, ASA 3 patients had a low average DASI score of 23, well below the threshold of 34 reported to identify patients with reduced activity. Data collected from the WD were concordant with those from the 6MWT when considering clinical scales. Patients with a reduced activity as defined via their daily steps had lower MET (4 vs. 6), lower DASI scores (30 vs. 40), and higher frailty (3.5 vs. 2) compared to standard activity patients. As clinical scales are the simplest and most pervasive criteria for objective preoperative evaluation, we propose that future research investigates the link between the preoperative and frailty scale, and patient home activity and mobility measured by the WD. This is crucial for validating and improving future WDs as a low-cost healthcare intervention.

Minutes of physical activity were divided into classes based on activity intensity. The trend of most patients confirms a sedentary life in this aged population, with most patients performing only light physical activity, such as simple house management and everyday life tasks. Some patients reported that during the perioperative period, they were less likely to exercise and take part in activities outside their household, partially because of stress and anxiety that led to social withdrawal and partially because of frequent hospital visits and waiting periods that caused a significant change in their routine.

VO2 max measurement is considered the goal standard for the assessment of cardiovascular fitness [13], however VO2 max is usually calculated during cycle ergometer testing at the maximal exercise capacity for the patient. This test is expensive and resource consuming in settings and is generally reserved for selected cases. In previous studies, investigators reported good estimations for physical activity and 𝑉𝑂2𝑚𝑎𝑥 in young and healthy subjects, while others investigated the role of step count and energy expenditure, highlighting the correlation with VO2 assessed using gold-standard methods. [19] Similarly, in this study, we tried to estimate the correlation between home physical performance, 𝑉𝑂2𝑚𝑎𝑥, and results from the 6MWT. In our population, many patients performed poorly for the calculation of this parameter (which is derived through a proprietary algorithm utilized by the Fitbit). The estimation of 𝑉𝑂2𝑚𝑎𝑥 was indeed computed daily by the Fitbit, but the data recording was not continuous. In fact, only 21 patients presented the value for at least one day, while the others had either no data or the value −1 (to mark missing data). Even if our results seem to align with the fact that a higher level of VO2 max relates to better patient fitness, the lack of data about some of the most fit patients prevents an assessment between these patients and their 𝑉𝑂2𝑚𝑎𝑥 values. On the contrary, patients that were considered the least fit in this population showed the lowest levels of VO2 max, with good correlation. The possibility to determine 𝑉𝑂2𝑚𝑎𝑥 could prove to be an interesting application of wearable devices in preoperative testing, if further developed by manufacturers and validated in clinical practice.

Most patients had similar values of respiratory rate before and after the 6MWT, partially related to the nature of the test, which is self-paced. Patients with the largest variations often presented significant variations in the Borg dyspnoea scale and were overall less fit than average or presented some form of pulmonary disease such as COPD. Similarly, heart rate variations between the resting and the maximal heart rate during the 6MWT were on average considerable (73 bpm versus 91 bpm); still, they remained predominantly in the normal range. Heart rate measured by the device’s PPG sensor showed a significant difference among patients. Several patients reached a HR greater than 160 bpm, while others rarely surpassed 120 bpm, and we found no correlation between HR and the clinical scales or the 6MWT results.

More than half of subjects presented an ASA score of II, that depicts patients with mild symptoms of disease who do not have significant impairment of functions. In this group, no patient suffered from complications. ASA III patients, which describes individuals with at least one severe disease, accounted for 32.3% of cases. Two ASA III patients suffered from complications. One patient had multiple surgical complications with reinterventions but recovered and was discharged home. A second patient died after hospital discharge and readmission because of long-term complications linked to the extensive nature of their underlying disease, despite good preoperative evaluation scores. We found no association between complications and the preoperative evaluation scales, 6MWT results, or WD data. Given the low number of subjects involved in this pilot study, our study was insufficient in detecting an association between complications and the clinical scales in these populations. 

Height and weight had no correlation with the 6MWT distance, while the body mass index had a mild negative correlation of −0.35, confirming that a larger body mass weight has a negative influence on the ability of patients to walk.

### 4.1. Technical Aspects of Commercial Wearable Devices

The Fitbit Inspire 2 is a basic activity tracker; more sophisticated models, available on the market, could allow a broader and more accurate recording of health parameters. However, the object of the investigators was to demonstrate that even data from an affordable device could be used to track physical activity, and thus act as a proxy of cardiopulmonary function. These commercial devices have two major theoretical advantages over more complex and expensive devices, including medical devices or high-range trackers. On the one hand, they are more widespread, specifically in younger people performing sport and other activities, and thus this or other similar commercial devices are already present in a large part of the population, constituting a mine of data that could be used in the future for health purposes, prehabilitation, and the personalization of health care recommendations. On the other hand, in the perspective of national health systems, a low-cost device which is reusable is more affordable and convenient, easier to buy, and can compete with the costs of the 6MWT. Even if a formal cost-analysis was not conducted in this study, a single 6MWT test has a cost comparable to the cost of these trackers, which is around USD 70–90, depending on vendors and time variations in prices. A single WD could also be re-used several times, after standard disinfection, further reducing the costs. The use of affordable commercial devices could be especially advantageous in low resource settings, such as in rural populations or Low and Medium Income (LMIC) countries, where the cost of medical devices or high-end commercial health-bands could be unsustainable [16]. High-end devices are able to collect more data, including blood oxygen saturation, but their initial cost can be two or three times higher compared to our devices [22]. Beside the higher costs of these devices, we can mention another two limits of high range devices: firstly, they are generally larger and heavier to wear, and this may reduce patient compliance in wearing them for several days. Secondly, more complex functions reduce battery life, thus requiring the patient to individually charge the device, a procedure which can be an obstacle for elderly patients. In our study, the battery life of the device provided did not need a recharge during the week of data collection. The ability to track simple heart, mobility, and energy expenditure data, as provided by low-end devices, could for these reasons be a good method for assessing patients’ fitness in both low and intermediate resource settings.

Moreover, in the years of the COVID-19 pandemic, when access to hospitals was greatly reduced and online preoperative evaluations were proposed and recommended [23], these type of devices which allow exercise testing without hospital access could be further advantageous compared to standard in-hospital exercise testing.

The accuracy of the data given by the device of choice, the Fitbit Inspire 2, has not been extensively discussed in the medical literature. Most of the studies comparing the accuracy of measure from a Fitbit wearable with a clinical-grade measure used different models, such as the Fitbit Charge, Surge, or Versa, which are able to collect more data [22,24]. In the future, more sophisticated devices, able to record ECG, SpO2, perfusion, or other variables, could be used to stratify patients for preoperative evaluation. Moreover, the gathering of longitudinal wearable data of physical activity could be useful for epidemiological studies and the quick determination of accurate normal ranges for health parameters. WD data could also be used in the identification of relevant disease before symptoms onset in acute [25] or chronic pathologies, in which a WD could also be used for monitoring. To ensure a faster implementation of wearable devices in healthcare, grey areas in legislation should be eviscerated to ensure a fair competition in the market and the safety and security of the individuals and privacy. This could also allow for better cooperation among developers, researchers, healthcare professionals, and regulatory institutions to develop opportunities towards the application of wearable technologies. The employment of machine and deep learning models (ML and DL) would also be applicable in this setting in order to discover new features and investigate different aspects of the collected data. 

### 4.2. Limitations

This was a pilot study, performed to assess whether wearable devices are a feasible proxy of preoperative clinical scales and tests. Accordingly, the number of patients included is low, and the study is insufficient in identifying patients at risk of complications. Nonetheless, we were able to demonstrate associations between the WD data, preoperative evaluation scales, and 6MWT results, which are designed to identify patients at risk of complications.

The collected data presented some inconsistencies for a specific number of parameters; the HR data had considerable variations in the recording, even for continuous logging. The logging should have been between 5 and 15 s; however, the data were often missing for longer periods. This aspect impacted on the possibility of recording HR variability using Detrended Fluctuation Analysis and respiratory rate derived using the Inter-beat Interval analysis, since short-term fluctuations are not computable with the lack of refined data provided by the device employed in this study.

The completeness of data has been an issue regarding several domains. HR and walked steps during the performance of the 6MWT were often missing or unreliable, even if the Fitbit was given to the patient before the beginning of the test. Jumps of different lengths were found often in data strings, with variable lengths, and they were probably due to motion artifacts which are common in wrist-worn devices [15]. However, it was not possible to verify the actual cause of data loss at the beginning of the 6MWT, which can also be related with a different sampling rate when the device is worn for the first time by a patient, since it was not possible to access raw sensor data. This limited the possibility of comparing the WD data during the 6MWT with those collected by clinical operators during the test.

## 5. Conclusions

We present the first study on the use of a commercial, wearable device as a proxy of the clinical and cardiopulmonary function test in the preoperative evaluation. Wearable devices demonstrated good correlation with the 6-minute walking test and with validated preoperative clinical evaluation scales. This pilot study represents a starting point for the implementation of a functional capacity in the preoperative evaluation, using low-cost, consumer-grade wearable technology. Further studies are needed to assess and validate the role of wearable devices in the perioperative setting.

## Figures and Tables

**Figure 1 diagnostics-13-00630-f001:**
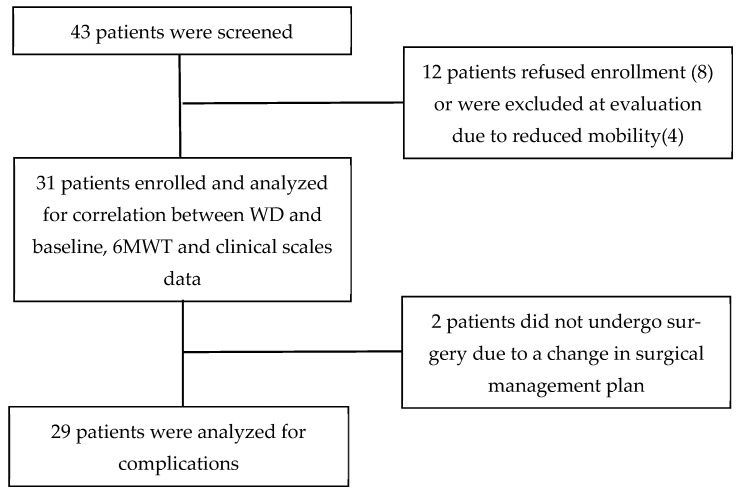
Flowchart of study process.

**Figure 2 diagnostics-13-00630-f002:**
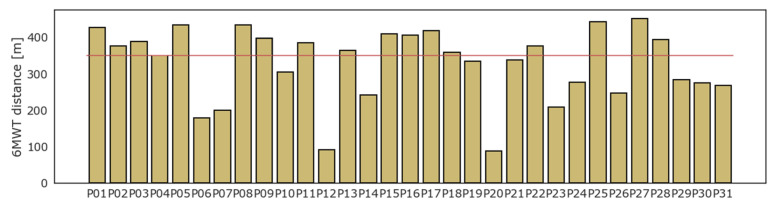
Six minutes walking test: results in distance (meters).

**Figure 3 diagnostics-13-00630-f003:**
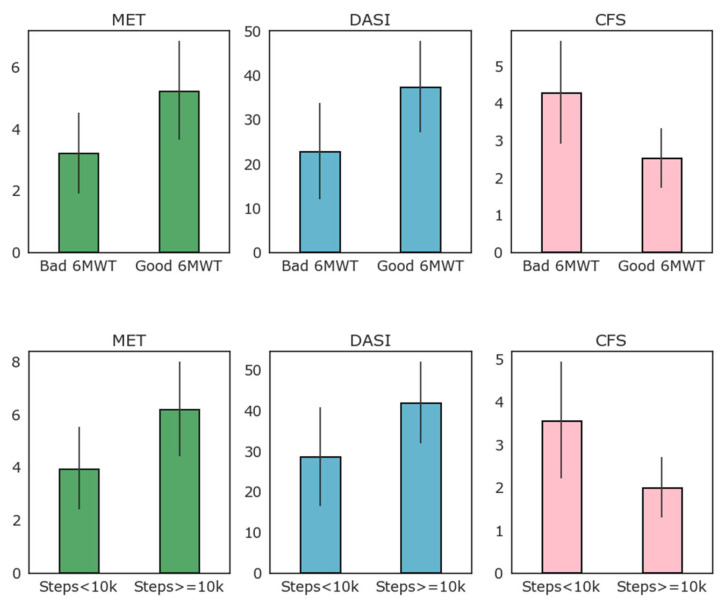
Relation of clinical evaluation scales to low–high performance 6MWT and to high–low average number of steps per day (above or below 10,000 steps per day on average).

**Figure 4 diagnostics-13-00630-f004:**
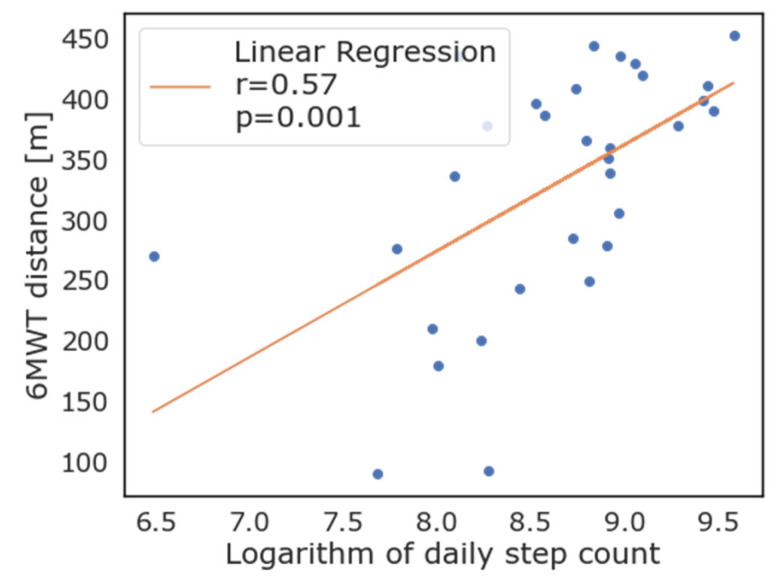
Correlation between logarithmic transformation of daily steps as measured by wearable devices and 6-minute walking test.

**Figure 5 diagnostics-13-00630-f005:**
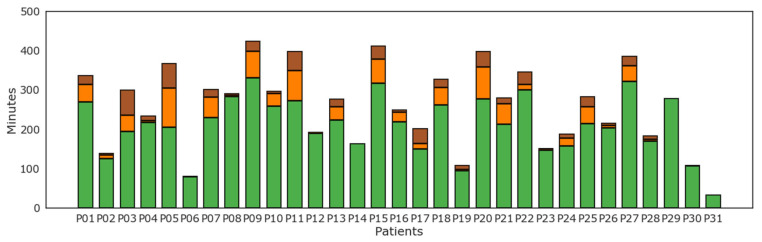
Minutes of physical activity according to level of intensity of physical activity: lightly active (green), moderately active (orange), and very active (brown) minutes for each patient (P01–P31).

**Figure 6 diagnostics-13-00630-f006:**
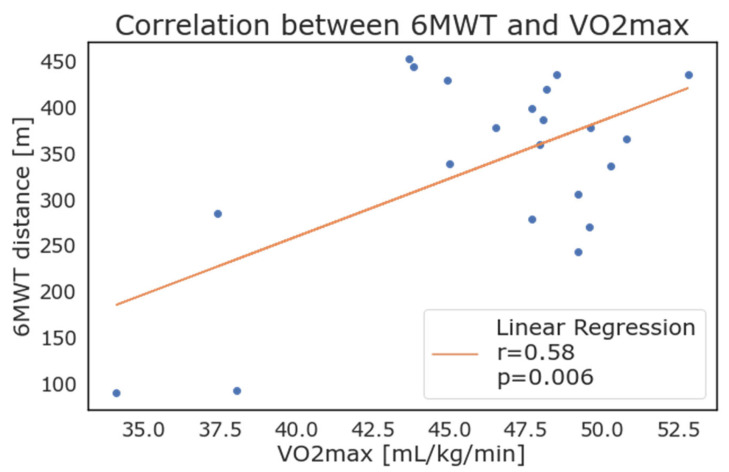
Correlation between 6-minute walking test results and maximal oxygen consumption (VO2max) as calculated and reported by wearable devices.

**Table 1 diagnostics-13-00630-t001:** Preoperative data, according to ASA class. BMI = Body Mass Weight, CFS = Clinical Frailty Scale, MET = Metabolic equivalent of tasks, DASI = Duke Activity Status Index.

	Total (*n* = 31)	ASA 1-2 (*n* = 20)	ASA 3-4 (*n* = 11)
Age (years)	76.1 ± 4.9	75.9 ± 3.62	78.97 ± 5.91
Height (cm)	167.9 ± 10.6	167.3 ±10.4	169.2 (11.4)
Weight (Kg)	70.4 ± 12.7	71.1 ± 12.6	69.3 ± 13.3
BMI (Kg/m^2^)	24.8 ± 2.8	25.3 ± 3.2	23.9 ±1.9
Male	22 (71.0%)	14 (70.0%)	8 (72.7%)
Smoker	18 (58.1%)	13 (65.0%)	5 (45.5%)
CFS score	3.32 ± 1.40	2.90 ± 1.07	4.09 ± 1.64
MET score	4.32 ± 1.78	4.75 ± 1.74	3.55 ± 1.63
DASI	30.82 ± 12.73	34.79 ± 9.94	23.60 (14.47)
Region			
Abdomen	9 (29.0%)	5 (25.0%)	4 (36.4%)
Liver	6 (19.4%)	5 (25.0%)	1 (9.1%)
Lung	7 (22.6%)	3 (15.0%)	4 (36.4%)
Urological	5 (16.1%)	4 (20.0%)	1 (9.1%)
Other	4 (12.9%)	3 (15.0%)	1 (9.1%)

**Table 2 diagnostics-13-00630-t002:** Total daily steps, 6MWT performance data, preoperative evaluation scales, and patient characteristics according to 6MWT threshold and to WD daily steps threshold.

	Total (*n* = 31)	6MWT < 350 m (*n* = 14)	6MWT > 350m (*n* = 17)	*p*-Value *	Daily Steps < 10,000 (*n* = 16)	Daily Steps > 10,000 (*n* = 15)	*p*-Value *
6MWT distance (m)	328.94 ± 99.53	239.79 ± 78.12	402.35 ± 30.56	<0.001	270.56 ± 104.34	391.20 ±38.90	<0.001
ASA 1-2	20 (65%)	7 (50%)	13 (76%)		9 (56%)	11 (73%)	
ASA 3-4	11 (35%)	7 (50%)	4 (24%)	0.2	7 (44%)	4 (27%)	0.3
Borg Scale pre-test	0.32 ± 0.79	0.64 ±1.08	0.06 ± 0.24	0.016	0.31 ± 0.48	0.33 ± 1.05	0.3
Borg Scale post-test	2.60 ± 2.09	3.14 ± 2.18	2.15 ± 1.97	0.2	2.75 ± 1.91	2.43 ± 2.32	0.4
HR pre-test	73.03 ± 13.00	75.50 ± 11.35	71.00 ± 14.23	0.2	74.00 ± 12.92	72.00 ± 13.45	0.5
HR post-test	81.37 ± 17.90	84.40 ± 15.81	79.59 ± 19.25	0.4	84.00 ± 16.13	79.27 ± 19.48	0.4
SpO2 pre-test	97.16 ± 1.19	96.79 ± 1.37	97.47 ± 0.94	0.2	97.12 ± 1.41	97.20 ± 0.94	>0.9
SpO2 post-test	94.52 ± 2.50	94.30 ± 2.21	94.65 ± 2.71	0.5	95.17 ± 2.41	94.00 ± 2.54	0.3
RR pre-test	13.71 ± 1.78	13.85 ± 1.23	13.57 ± 2.24	0.95	13.73 ± 1.28	13.69 ± 2.29	0.5
RR post-test	17.14 ± 3.05	17.85 ± 2.53	16.42 ± 3.43	0.87	17.60 ± 2.64	16.62 ± 3.50	0.5
Complications	3/29 (10%)	1/12 (8.3%)	2/17 (12%)	>0.9	1/15 (6.7%)	2/14 (14%)	0.6
MET scale	4.32 ± 1.78	3.21 ± 1.31	5.24 ± 1.60	<0.001	3.75 ± 1.65	4.93 ± 1.75	0.079
CFS	3.32 ± 1.40	4.29 ± 1.38	2.53 ± 0.80	<0.001	3.88 ± 1.54	2.73 ± 0.96	0.051
DASI	30.82 ± 12.73	22.83 ± 10.83	37.39 ± 10.34	0.002	25.18 ± 11.72	36.84 ± 11.18	0.010

* Wilcoxon rank-sum test or Chi-Square test; 6MWT = 6-minute walking test, CFS = Clinical Frailty Scale, MET = Metabolic equivalent of tasks, DASI = Duke Activity Status index, RR: respiratory rate, HR = heart Rate, SpO2: peripheral oxygen saturation levels.

## Data Availability

Data are available from authors upon reasonable request.

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
