# Peer review of "Wearable Health Technology for Preoperative Risk Assessment in Elderly Patients: The WELCOME Study"

_diagnostics, 2023, doi:10.3390/diagnostics13040630_

Round 1

Reviewer 1 Report

Greco et al performed pilot study using wearable devices as a proxy for clinical and cardiopulmonary function using 6MWT. The following concerns needs to be addressed.

1. In the market, health bands are available at lower and higher cost compared to fitbit, can authors comment on the use of more cheaper and reliable WD for rural population which cannot afford fitbit inspire 2.

2. Author should compare with results with their WD and clinical tool, these will be helpful for researchers to improve the wearable devices. 

3. Authors should provide abbreviation of 'ASA', this will help to readers.

4. In page 8, line 283, 'patent' should be changed to 'patient' and page 9 line 351, 'choiche' changed to 'choice'

Reviewer 2 Report

The authors present an interesting study as a group. There are a few suggestions and revisions required in the manuscript

1. 6MWT - cannot be a keyword. Kindly use 5 keywords relevant to the topic. Choose keywords listed in the US National Library of Medicine’s collection of Medical Subject Headings (MeSH)

2. The ending note of the introduction section is not satisfactory. Lines 64-71 need major revision and rewriting. Kindly avoid using words as we investigated, we used, etc. Instead, use "in the present study"

3. The authors have split the content in section materials and methods into short paragraphs of 2 lines each (72-92). is better if it's clubbed and presented as a paragraph

4. Angelucci A et al should be Angelucci A et al. [] Kindly cite the reference immediately.

5. Figure 2. Figure 3, Figure 5., and Figure 6. images of high resolution with white backgrounds should be provided.

6. Detailed explanation of the observations and outcomes of Figure 4., Figure 5., and Figure 6. are required to be added.

7. Authors can consider further strengthening the introduction section with more recent studies like 

https://doi.org/10.3390/jcm11237019 

https://doi.org/10.1039/D2NR04551F

https://doi.org/10.1016/j.bja.2022.02.034

https://doi.org/10.1007/978-3-031-19647-8_20

https://doi.org/10.1021/acs.analchem.2c02642

https://doi.org/10.3390/healthcare10020275

Reviewer 3 Report

This paper studied the relation between the data from commercial wearable device to the standard preoperative assessment. It is very meaningful study.

The experiment arrangement is reasonable. But the structure of the paper needs to be revised before public. The suggestions are as follows:

1. Reduce the number of paragraphs. 

2. Merge discussion and results. Place the contents of the discussion in the appropriate placed in the results.

3. Improve the quality of Figures. Enlarge the letters on Fig. 5.
